# PGRP-LB: An Inside View into the Mechanism of the Amidase Reaction

**DOI:** 10.3390/ijms22094957

**Published:** 2021-05-07

**Authors:** Julien Orlans, Carole Vincent-Monegat, Isabelle Rahioui, Catherine Sivignon, Agata Butryn, Laurent Soulère, Anna Zaidman-Remy, Allen M. Orville, Abdelaziz Heddi, Pierre Aller, Pedro Da Silva

**Affiliations:** 1Univ Lyon, INSA Lyon, INRAE, BF2I, UMR 203, 69621 Villeurbanne, France; julien.orlans@insa-lyon.fr (J.O.); carole.monegat@insa-lyon.fr (C.V.-M.); isabelle.rahioui@inrae.fr (I.R.); catherine.sivignon@inrae.fr (C.S.); anna.zaidman@insa-lyon.fr (A.Z.-R.); abdelaziz.heddi@insa-lyon.fr (A.H.); 2Diamond Light Source, Harwell Science and Innovation Campus, Didcot, Oxfordshire OX11 0DE, UK; agata.butryn@diamond.ac.uk (A.B.); allen.orville@diamond.ac.uk (A.M.O.); 3Research Complex at Harwell, Rutherford Appleton Laboratory, Didcot, Oxfordshire OX11 0FA, UK; 4Univ Lyon, INSA Lyon, Université Claude Bernard Lyon 1, CPE Lyon, UMR 5246, CNRS, ICBMS, Institut de Chimie et de Biochimie Moléculaires et Supramoléculaires, Bât. E. Lederer, 1 rue Victor Grignard, 69622 Villeurbanne, France; laurent.soulere@insa-lyon.fr

**Keywords:** peptidoglycan recognition protein, PGRP-LB, X-ray crystallography, *Drosophila melanogaster*, innate immunity

## Abstract

Peptidoglycan recognition proteins (PGRPs) are ubiquitous among animals and play pivotal functions in insect immunity. Non-catalytic PGRPs are involved in the activation of immune pathways by binding to the peptidoglycan (PGN), whereas amidase PGRPs are capable of cleaving the PGN into non-immunogenic compounds. *Drosophila* PGRP-LB belongs to the amidase PGRPs and downregulates the immune deficiency (IMD) pathway by cleaving *meso*-2,6-diaminopimelic (*meso*-DAP or DAP)-type PGN. While the recognition process is well analyzed for the non-catalytic PGRPs, little is known about the enzymatic mechanism for the amidase PGRPs, despite their essential function in immune homeostasis. Here, we analyzed the specific activity of different isoforms of *Drosophila* PGRP-LB towards various PGN substrates to understand their specificity and role in *Drosophila* immunity. We show that these isoforms have similar activity towards the different compounds. To analyze the mechanism of the amidase activity, we performed site directed mutagenesis and solved the X-ray structures of wild-type *Drosophila* PGRP-LB and its mutants, with one of these structures presenting a protein complexed with the tracheal cytotoxin (TCT), a muropeptide derived from the PGN. Only the Y78F mutation abolished the PGN cleavage while other mutations reduced the activity solely. Together, our findings suggest the dynamic role of the residue Y78 in the amidase mechanism by nucleophilic attack through a water molecule to the carbonyl group of the amide function destabilized by Zn^2+^.

## 1. Introduction

Peptidoglycan (PGN) is a major bacterial cell wall component that allows the conservation of the cell shape and prevents the bursting of the bacteria [1,2]. PGN is a polymer composed of linear glycan strands that are cross-linked by short peptides. The glycan strand is made of alternating β-1,4-connected N-acetylglucosamine (GlcNAc) and N-acetylmuramic acid (MurNAc) residues. The peptide stem varies among species but is usually a pentapeptide and contains L- and D-amino acids. The third residue can be either a *meso*-2,6-diaminopimelic acid (*meso*-DAP or DAP) for Gram-negative bacteria and Gram-positive bacilli or a L-lysine (L-Lys) for other Gram-positive bacteria [1,2]. PGN, and its associated muropeptides like the tracheal cytotoxin (TCT), belong to the microbial-associated molecular patterns (MAMPs), which interact with the host pattern recognition receptors (PRRs) and activate the host innate immune system [3,4,5]. Some of these PRRs are peptidoglycan recognition proteins (PGRPs) that are ubiquitous in most animals and play a pivotal role in the innate immune system.

The first PGRP was described in the silkworm *Bombyx mori* [6]; further studies showed that PGRPs are widespread among invertebrates and vertebrates. However, this gene family has not been found in plants or in lower metazoan, such as nematodes, so far [7]. Insect PGRPs were shown to be more diverse when compared to other species [7]. The fruit fly *Drosophila melanogaster* genome contains 13 PGRP genes encoding over 20 different proteins [8,9] that are involved in triggering the two major antibacterial immune signaling pathways: the immune deficiency (IMD) and Toll pathways. Distinct PGRPs are specific to DAP-type or Lys-type PGN, leading to the activation of IMD or Toll pathways, respectively. Both pathways result in the synthesis and secretion of antimicrobial peptides (AMPs) that are directed against microbial intruders [10]. In *Drosophila*, PGRP-LC and PGRP-LE recognize DAP-type PGN and activate the IMD pathway [11], while PGRP-SA recognizes Lys-type PGN and activates the Toll pathway [12].

Other PGRPs, including *Drosophila* PGRP-LB [13] and PGRP-SB1 [14], belong to the N-acetylmuramoyl-L-alanine amidase and cleave PGN into non-immunogenic compounds. Amidase PGRPs are also found in mammals, for example, the human PGLYRP2 [15]. *Drosophila* PGRP-LB is specific to the DAP-type PGN. Its expression is regulated at the transcriptional level by the IMD pathway. This IMD-regulated expression of PGRP-LB in the gut avoids a constitutive local immune response to microbiota bacteria, and creates a threshold of infection for a systemic immune response activation [13,16]. The *Drosophila* PGRP-LB gene encodes three isoforms, two cytosolic (PGRP-LB^PA^ and PGRP-LB^PD^) and one secreted (PGRP-LB^PC^) [17]. The secreted isoform PGRP-LB^PC^ and cytoplasmic isoform PGRP-LB^PA^ are the same protein (PGRP-LB^PA/PC^), and following the signal peptide of PGRP-LB^PC^, they share the same sequence (Figure 1A). The second cytoplasmic isoform, PGRP-LB^PD^, displays a longer N-terminal than PGRP-LB^PA/PC^ with no homology or known function (Figure 1A). It was proposed that PGRP-LB^PC^ cleaves the PGN in the gut lumen, avoiding a constitutive and systemic immune response to gut microbiota, while PGRP-LB^PA^ and PGRP-LB^PD^ have a redundant role in degrading PGN in enterocytes [17]. In insects coevolving with intracellular symbiotic bacteria (endosymbionts), the PGRP-LB protein was shown to downregulate the immune response against endosymbionts in the tsetse fly [18,19], and to maintain host homeostasis by cleaving endosymbiont TCT in the cereal weevils [20].

Structural studies determined that amidase and non-catalytic PGRPs share the same structural domain involved in PGN recognition [21,22,23,24,25,26,27,28,29]. The structure of the PGRP domain comprises three α-helices and one central β-sheet composed of six β-strands, similar to the bacteriophage T7 lysozyme, a zinc-dependent amidase [21,30]. The main difference between amidase and non-catalytic PGRPs is the presence of Zn^2+^ in only amidase PGRPs, which is coordinated by two histidines and one cysteine. Previous non-catalytic PGRP structures have provided a more comprehensive view on the residues responsible for the specificity towards the two types of PGN (Figure 1B). For the analysis of DAP-type specificity, the TCT has been complexed to *Drosophila* PGRP-LC [23] or *Drosophila* PGRP-LE [24]. As for the Lys-type specificity, human PGLYRP3 has been complexed to muramyl tripeptide (MTP) [26] or muramyl pentapeptide (MPP) [27], and glucosamyl muramyl pentapeptide (GMPP) has been complexed to human PGLYRP4 [28]. However, the N-acetylmuramoyl-L-alanine amidase reaction mechanism remains unclear because of the lack of an amidase PGRP structure in interaction with a PGN compound. So far, only protein structure superposition and ligand docking have been used to propose a mechanism [26].

In the present study, we analyzed the specific enzymatic activity of different *Drosophila* PGRP-LB isoforms towards various PGN substrates to understand their specificity and role in the immune system of the fruit fly. All isoforms presented similar activity towards different compounds. We generated mutants to characterize the amidase reaction mechanism and solved the X-ray structures of wild-type *Drosophila* PGRP-LB^PA/PC^ and its mutants in apo form or complexed to TCT. The analysis of these results, including the first structure of a catalytic PGRP associated with TCT, highlighted the dynamic role of the residue Y78 in the amidase activity. Altogether, our results lead to a new reaction mechanism for the N-acetylmuramoyl-L-alanine amidase activity that could be applied to all the amidase PGRPs.

## 2. Results and Discussion

### 2.1. Drosophila PGRP-LB Isoforms Have a Similar Amidase Activity

*Drosophila* PGRP-LB has a specific activity of cleaving DAP-type PGN and its muropeptides into non-immunogenic compounds [13,16]. More recently, three isoforms of this protein have been described with different tissue localization and pattern of expression [17]. The activity of these isoforms against muropeptides with varying sugar moieties has never been characterized. We hypothesized that the longer N-terminal of PGRP-LB^PD^ could be responsible for giving to the enzyme a specific activity towards some substrates compared to the PGRP-LB^PA/PC^ isoforms. In order to precisely define the substrate specificity of the PGRP-LB^PA/PC^ and PGRP-LB^PD^ isoforms, a series of polymeric and monomeric PGN substrates were tested using an in vitro enzymatic activity assay (Table 1 and Appendix A). Both PGRP-LB^PA/PC^ and PGRP-LB^PD^ isoforms displayed the same, and the strongest activity on the GM(anh)-tetra_DAP_ (also called TCT) substrate while remaining very active against other muropeptides and polymeric DAP-type PGN. Therefore, the presence of the longer N-terminal in PGRP-LB^PD^ did not seem to induce any specificity for a particular compound leading to any influence in activity.

This extra sequence could bring a supplementary domain complementing the protein activity of PGN degradation, similar to the PGRP3 from *Branchiostoma belcheri tsingtauense,* where the PGRP domain is fused with a chitin binding domain [32]. However, this N-terminal does not show any homology and its function could not be predicted. This is why, in the rest of the article, the research will focus on the PGRP-LB^PA/PC^ isoform, described as PGRP-LB in previous papers [13,21].

### 2.2. Identification of Potential Key Residues in PGRP-LB for Amidase Reaction

Structural analysis of *Drosophila* PGRP-LB [21] and T7 Lysozyme [30], along with a multiple sequence alignment of PGRPs from different organisms, reveal the highly conserved residues of *Drosophila* PGRP-LB^PA/PC^ histidine 42 (H42), histidine 152 (H152), tyrosine 78 (Y78) and cysteine 160 (C160) in coordination with Zn^2+^ in the catalytic site (Figure 1B and Figure 2A). Y78 residue is invariant among amidase and even non-catalytic PGRP sequences from different organisms (Figure 1B). In the T7 Lysozyme, mutation of the conserved tyrosine residue to a phenylalanine (Y46F), corresponding to Y78 in *Drosophila*, results in a loss of the amidase activity [30]. This mutation highlights the crucial role played by the conserved tyrosine in the enzymatic process. The H42, H152 and C160 residues in *Drosophila* PGRP-LB^PA/PC^ sequence are conserved only in the catalytic PGRPs (Figure 1B).

Because no other residues seemed close enough to be involved the catalytic reaction or in the stabilization of the transition state during the enzymatic reaction, we raised two alternative hypotheses: (a) one of the Zn^2+^ chelating histidines is also involved in the stabilization of the transition state, or (b) only the tyrosine is involved in the amidase reaction and no stabilization of the transition state is required.

To tackle these two scenarios, we designed and analyzed four different mutations of the conserved residues in the catalytic *Drosophila* PRGP-LB^PA/PC^: H42A, Y78F, H152A and C160S.

### 2.3. Only Y78 Residue Is Necessary for the Amidase Reaction

We performed enzymatic activity assays on the four selected mutants, using the same compounds as for the isoforms of *Drosophila* PGRP-LB (Table 1 and Appendix A). The activity of PGRP-LB^PA/PC^_C160S was firstly described as abolished [13]. Our results show, however, that the activity of the H42A, H152A and C160S PGRP-LB^PA/PC^ mutants depends on the substrate type and is either significantly reduced or completely abolished. Only the Y78F mutation caused complete loss of the enzymatic activity for every compounds (Table 1). To rule out the possibility that the reduced activity of the mutants is caused by impaired recognition of the PGN substrate, we performed binding assays of H42A, Y78F, H152A and C160S mutants as well as the wild-type PGRP-LB^PA/PC^ with a DAP-type polymeric PGN. The introduced mutations did not affect the ability to recognize PGN (Appendix A), attesting that Y78 is the only residue taking part in the amidase mechanism.

### 2.4. C160S Is the Most Essential Residue for Zn^2+^ Chelation

The structure of the *Drosophila* PGRP-LB^PA/PC^ wild-type was solved by X-ray crystallography (PDB 7NSX) (Table 2). The resolution was slightly improved (1.90 Å) from the previous publication (PDB 1OHT, 2.00 Å) [21]. These two structures are almost identical with a calculated RMSD on the Cα of 0.90 Å and obtained in different space groups, P6_1_22 for 1OHT and C222_1_ for 7NSX. In both structures, the Zn^2+^ is coordinated by H42, H152 and C160. Yet, the water molecule involved in the interaction between the residue Y78 and Zn^2+^ is replaced by a residue of the C-terminal belonging to a symmetry-related molecule, E182 and D180 respectively in 1OHT and in 7NSX (Appendix A). This interaction is non-physiological and can be considered as an artefact of crystallization between the C-terminal residues of a symmetry related molecule and the active site. In solution, one should expect that a water molecule would interact with both the residue Y78 and Zn^2+^ like in the T7 Lysozyme structure (Y46 and Zn^2+^) [30].

The structures of the mutants C160S and Y78F PGRP-LB^PA/PC^ were solved by X-ray crystallography (respectively, PDB 7NSY and PDB 7NSZ) (Table 2). Unfortunately, we were unable to crystallize H42A and H152A PGRP-LB^PA/PC^ mutants. Y78F and C160S PGRP-LB^PA/PC^ mutant structures were elucidated to 1.30 Å and 1.40 Å resolution, respectively. Both mutants Y78F and C160S PGRP-LB^PA/PC^ share a similar fold with PGRP-LB^PA/PC^ wild type with a RMSD on Cα of 0.49 Å and 1.12 Å, respectively, but in different space groups, P6_1_22 for PGRP-LB^PA/PC^_Y78F and P1 for PGRP-LB^PA/PC^_C160S (Figure 2B–D). PGRP-LB^PA/PC^_Y78F also presents a crystal packing artefact where E182 from the C-terminal of a symmetry-related molecule makes a coordination with Zn^2+^ (Appendix A). However, major structural changes were observed in PGRP-LB^PA/PC^_C160S structure around the mutation. The loop between the β6 sheet and the α3 helix is drifting apart from the active site with a RMSD of 2.37 Å, calculated on the Cα from the residues 148 to 172. This deviation is due to the loss of interaction between C160 and Zn^2+^, where the cysteine is replaced by a serine (Figure 2C). Additionally, the presence or absence of Zn^2+^ was assessed by X-ray emission fluorescence on PGRP-LB^PA/PC^ wild type, Y78F and C160S crystals (Figure 2E–G). We showed that Zn^2+^ is totally absent in PGRP-LB^PA/PC^_C160S mutant as already observed in the structure resolution (Figure 2C), while it is still present in PGRP-LB^PA/PC^_Y78A mutant and PGRP-LB^PA/PC^ wild type. Noticeably, no Zn^2+^ was added to the buffer during purification and crystallization steps, meaning that the Zn^2+^ present in the active site of PGRP-LB^PA/PC^_Y78A and PGRP-LB^PA/PC^ proteins comes from the expression cell system.

Finally, the addition of Zn^2+^ to the reaction buffer used for activity assays (Table 1) allowed a small residual amidase activity of PGRP-LB^PA/PC^_C160S, whereas the absence of Zn^2+^ in the buffer totally abolished the amidase activity of PGRP-LB^PA/PC^_C160S, which is consistent with X-ray structural data (Figure 2C,F). These data suggest that the C160 residue is needed to retain Zn^2+^ in the active site but not essential for residual amidase activity if Zn^2+^ is present in the media. For the H42A and H152A PGRP-LB^PA/PC^ mutants, they remain active even without any Zn^2+^ added to the reaction buffer, meaning that Zn^2+^ could be still partially present in the active site. Regarding PGRP-LB^PA/PC^_Y78F, the amidase activity is also abolished without Zn^2+^ in the buffer. Of note, in the wild-type PGRP-LB^PA/PC^ and PGRP-LB^PD^ even without any Zn^2+^ in the media, the activity remains the same confirming the strong chelation of Zn^2+^ by the protein.

### 2.5. TCT Makes Two Important Interactions with R92 and Zn^2+^

Initial X-ray and enzymatic analyses showed that the Y78F mutation does not destabilize the protein structure and prevents PGN substrate from being degraded. Hence, we used this mutation to study the interaction between the PGRP-LB^PA/PC^ and one of the muropeptides, the TCT with X-ray crystallography (PDB 7NT0) (Table 2). In this structure elucidated to 1.80 Å resolution, the ligand is buried in a mostly positively charged pocket. Whereas the tetra-peptide is well defined, the carbohydrate moiety of TCT is not visible in the electron density (Figure 3A).

The specificity of the PGRP-LB^PA/PC^ to the PGN is ensured by the recognition of the *meso*-DAP residue, interacting via hydrogen bonds with R92 (Figure 3B,C). Previous studies pointed out this interaction between the DAP residue and an arginine as observed in PGRP-LE [24] and PGRP-LC [23], and by comparing different DAP-type PGRP sequences, the arginine residue appears well conserved (Figure 1B). For the Lys-type specific PGRPs, this residue is replaced by a threonine in *Drosophila* PGRP-SA [29] and a valine in human PGLYRP3 [26,27] and PGLYRP4 [28] (Figure 1B). This specificity of *Drosophila* PGRP-LB^PA/PC^ for the DAP-type PGN is confirmed with an activity around 30 times higher for the TCT compared to the GM(anh)-penta_Lys_ (Table 1).

The structure obtained here also shows another interesting interaction with a fourth coordination of Zn^2+^ made with the oxygen from the amide group of the TCT, which highlights the important role of Zn^2+^ in the enzymatic reaction (Figure 3B,C). This interaction would increase the electrophilic character of the carbonyl group of the amide function where the amidase activity is occurring.

By comparing the apo and complexed PGRP-LB^PA/PC^_Y78F structures, the C-terminal symmetry-related molecule is filling the binding pocket of the peptide moiety from the PGN attesting the non-physiological nature of this interaction (Figure 3C,D).

### 2.6. PGRP-LC and PGRP-LE Helix α1 Allows a Stronger Recognition of the Sugar Moiety

The absence of the electron density for the carbohydrate moiety of the TCT in the PGRP-LB^PA/PC^ complexed structure is most probably due to its high flexibility. There are two possible pockets that can accept the sugar moiety of the ligand (annotated P1 and P2 in Figure 3A). The first pocket (P1) has a well-defined electron density corresponding to 5 residues of the N-terminal from a symmetry-related molecule (Figure 3B). This rearrangement, different from the apo structure, is also probably due to the crystal packing and may not be biologically relevant. The second pocket (P2) could accommodate the sugar moiety of the TCT. The structure of PGN recognition PGRPs, PGRP-PGRP-LE [24] and PGRP-LC [23], in complex with TCT, have been solved previously. In these structures, the electron density is well defined throughout the entire ligand and the sugar is located in the same pocket as the pocket (P2) in the PGRP-LB^PA/PC^_Y78F. We can then assume that the pocket (P2) is accommodating the sugar moiety even though no interpretable electron density is visible in the vicinity.

This PGRP-LB^PA/PC^ _Y78F structure complexed with TCT allows to better understand the PGN binding and multimerization event in the non-catalytic PGRPs in *Drosophila*. PGRP-LC and PGRP-LE are PGN receptors involved in the activation of the IMD pathway [11]. To activate the IMD pathway, these proteins form an amyloidal signaling complex through their RHIM motif [36,37]. In the formation of amyloids, PGRP-LC is organized in a heterodimer [23], while PGRP-LE in a homomultimer [24]. The dimer interface of these PGRPs is interacting with TCT and is located at the carbohydrate moiety of the muropeptide (Figure 4A,B). To better analyze the interaction of the sugar part of the TCT with the PGRP-LB, we have modelled the missing atoms in the pocket (P2) in the PGRP-LB^PA/PC^_Y78F + TCT structure (Figure 4C).

The majority of the residues from PGRP-LCa and PGRP-LE, responsible for the dimerization in the complexed structures, directed to the anhydro bond and the GlcNAc part of the TCT, are located in the helix α1 (Figure 1B). In the X-ray structure of PGRP-LB^PA/PC^, this dimerization involving the helix α1 was not observed. Moreover, in the model no residue was able to balance the missing hydrogen bond network stabilizing the sugar moiety explaining the absence of electron density (Figure 4C).

The sugar moiety of TCT is involved in stabilizing the dimerization via a strong hydrogen bond network (Figure 4A,B). A shorter sugar will make fewer interactions and will result in a weaker dimerization, which could explain why other muropeptides other than TCT can still activate the IMD pathway to a lesser extent [3]. As for PGRP-LB, its scavenger role only needs the degradation towards all PGN substrates and therefore the dimerization to increase the specificity is not necessary.

### 2.7. Amidase PGRP Reaction Mechanism Needs the Dynamic Role of Y78

Zn^2+^ is one of the most important cofactors, allowing metalloenzymes interacting with this ion to catalyze essential reactions [38]. Here, we demonstrate the essential role played by Zn^2+^ in *Drosophila* PGRP-LB^PA/PC^ and in N-acetylmuramoyl-L-alanine amidase in general. Our findings also highlight the pivotal role and major implication of the Y78 residue in the catalytic process of PGRP-LB.

During the analysis of the structure of PGRP-LB^PA/PC^_Y78F + TCT, the residue H67 seemed worthy of further investigation as it was previously suggested to be responsible for the transition-state stabilization [21]. It interacts with the nitrogen of the amide group between the sugar moiety and the peptide stem through an Asn of the N-terminal brought by the crystal packing (Figure 3C). In the PDB 1OHT structure, H67 forms a hydrogen bond with a water molecule, raising the question of its participation in the mechanism [21]. We tested the enzymatic activity of a mutant H67A against the *E. coli* polymeric PGN and the specific activity was 1041 ± 34 nmol.min^−1^.mg of protein^−1^, which indicates that the mutation does not change the enzyme activity and binding affinity to PGN. This result shows that H67 is not involved in the reaction mechanism.

Enzymatic assays and structural data suggest that the catalytic amidase PGPRP-LB proceeds through the chemical mechanism displayed in Figure 5. First, a nucleophilic attack of the water stabilized by the Y78 occurs. It is the interaction between the substrate and Zn^2+^ that increases the electrophilic character of the carbonyl group of the amide function, which allows this attack. We then have a formation of a tetrahedral intermediate stabilized by Y78, leading to a concerted rearrangement assisted by Y78, which results in the formation of two products. Our analysis points to a dynamic role of this tyrosine in the amidase mechanism from substrate binding to product release, in contrast with the previous mechanism proposed, which was based only on a model without experimental data [26], and did not consider enough the crucial and dynamic role of Y78. This analysis also shows how the mechanism involved with PGRP-LB differs from bacterial N-acetylmuramoyl-L-alanine amidases such as AmiD [39] and AmiA [40].

In conclusion, we have shown the redundant activity of the three isoforms of *Drosophila* PGRP-LB, all presenting a similar activity towards the polymeric PGN and its associated muropeptides. In addition to this observation, PGRP-LB presents a wide range of activity towards various DAP-type PGN compounds. It is capable of degrading into non-immunogenic compounds both *E. coli* polymeric PGN and muropeptides with variation on their sugar moiety, with a preference for the TCT. This result indicates that the sugar moiety is not as important as the DAP residue in the recognition and degradation of PGN by the *Drosophila* PGRP-LB. Moreover, we have characterized the pivotal role of the residue Y78 in the reaction mechanism of PGRP-LB and by extension in all amidase PGRPs. The tyrosine makes a nucleophilic attack through a water molecule to the carbonyl group of the amide function destabilized by a Zn^2+^ interaction. It is noteworthy that the three residues responsible for the Zn^2+^ chelation are not involved in the amidase reaction itself. This work provides a clearer view on the N-acetylmuramoyl-L-alanine amidase activity harbored by the catalytic PGRPs and the T7 Lysozyme.

## 3. Materials and Methods

### 3.1. Sequence Alignment

The different sequences of different PGRPs were gathered from the database UniPort [41]: PGRP-LB_Dm (Q8INK6), PGRP-SB1_Dm (Q70PY2), PGRP-LBi_Sz (A0A411IZR2), PGRP-LB_Gmm (Q2PQQ9), PGLYRP2_Hs (Q96PD5), T7-Lysozyme (P00806), PGRP-LE_Dm (Q9VXN9), PGRP-LCx_Dm (Q9GNK5), PGRP-SA_Dm (Q9VYX7), PGLYRP1_Hs (O75594), PGLYRP3_Hs (Q96LB9), PGLYRP4_Hs (Q96LB8). These sequences were aligned with the program Clustal Omega [42] and the result was displayed using ESPript3.0 [31].

### 3.2. Protein Cloning, Expression and Purification

The gene segment coding for the different isoforms of *Drosophila* PGRP-LB and its mutants were synthesized by gblock (Integrated DNA Technologies, Coralville, IA, USA). The genes were cloned into the pPOPINM plasmid (PPUK) [43] using NEBuilder HiFi DNA Assembly (NEB, Ipswich, MA, USA). The pOPINM plasmid encodes N-terminal His-MBP tagged proteins. Recombinant plasmids were transformed into *E. coli* DH5alpha (NEB, Ipswich, MA, USA) for plasmid amplification and subsequently into SHuffle^®^ T7 Express Competent *E. coli* cells (NEB, Ipswich, MA, USA) for protein expression. Cultured cells were grown in LB broth with ampicillin at 30 °C and induced at A_600_ of 0.8 by adding IPTG to a final concentration of 1 mM for overnight expression at 16 °C.

The bacterial pellet was lysed by sonication in 100 mM Trizma base pH 7.5, 500 mM NaCl and 20 mM Imidazole. The supernatant was then applied on a HisTrap HP 5 mL column (GE Healthcare, Chicago, IL, USA) and washed with the same buffer as the lysis. The protein was then eluted by increasing the imidazole up to 250 mM. The eluted fraction was overnight dialyzed in 100 mM Trizma base pH 7.5, 500 mM NaCl and 20 mM Imidazole with 3C Protease his-tagged, in order to remove the His-MBP tag. Further purification step was done by applying the protein again on the HisTrap column, the protein is then collected in the flow-through. Finally, a gel filtration is done on a HiPrep Sephacryl S-100 HR (GE Healthcare, Chicago, IL, USA) in 20 mM HEPES pH 7.5 and 150 mM NaCl. The proteins were concentrated with Amicon^®^ centrifugal concentrators (Merck Millipore, Burlington, MA, USA).

The protein purification quality was assessed by 4–12% SDS-PAGE (Appendix A).

### 3.3. Peptidoglycan and Muropeptides Purification

The preparations of the different PGN compounds were done as described before [3]. To sum up, the polymeric peptidoglycan was purified from *E. coli* mutant strain BW25113 Δ*6LDTs*::Kan^R^ that does not express all six L,D-transpeptidase genes. Culture was grown overnight at 37 °C in LB broth with kanamycin. The bacterial pellet was resuspended in a minimal volume of 0.9% NaCl and poured drop by drop in 100 mL of 4.5% SDS at 95–100 °C. After 1 h of incubation, the mixture was cooled down overnight at room temperature. The suspension was centrifuged for 20 min at 200,000× *g* and the pellet was washed several times with water. Finally, the peptidoglycan was resuspended in a minimal volume of ultrapure water. All the enzymatic reactions described in the Appendix A were performed overnight at 37 °C. The reaction is stopped by adding 2 µL of phosphoric acid per 500 µL of reaction. Before the injection in the HPLC, an equal volume of 50 mM sodium phosphate pH 4.5 was added to the reaction. The HPLC conditions and the retention times are described in Appendix A. After the first run of HPLC, the compounds were lyophilized and resuspended in ultrapure water for the second HPLC run. Finally, the compounds were lyophilized and resuspended in ultrapure water at the desired concentration.

The GM(anh)-penta_Lys_ compound was generated from the M. luteus PGN by following the same process.

The different purified compounds are represented in Appendix A.

### 3.4. Enzymatic Activity

The reaction mixture (50 μL) contains 20 mM HEPES pH 7.5, 1 mM ZnCl_2_, pure polymeric PGN or PGN fragment (0.2 mM) and the protein (0.02 to 5 μg, depending on the substrate and protein used). The reaction was left to incubate for 30 min at 37 °C, then 2 μL of phosphoric acid are added to stop the reaction. The decrease of substrate and the apparition of products were measured by HPLC to calculate the specific activity of the various proteins.

### 3.5. Interaction with PGN

The method was already described for the PGRP-LB [21]. Polymeric peptidoglycan from *E. coli* was incubated with PGRP-LB or its mutants in 20 mM HEPES pH 7.5 and 150 mM NaCl for 1 h at 4 °C. The samples were centrifuged at 16,000× *g* for 10 min and the supernatant fractions were collected representing the unbound fractions. PGN pellets were washed with the same buffer and then dissolved in Leammli buffer representing the bound fractions. The samples were then analyzed by 10% SDS-PAGE.

### 3.6. Protein Crystallization and Data Collection

The proteins were concentrated at 10 mg/mL in 20 mM HEPES pH 7.5 and 150 mM NaCl. For the TCT complexed structure, the muropeptide was added to a final concentration of 1 mM. The crystallization was performed by the sitting-drop vapor-diffusion method in 96-well CrystalQuick^TM^X plates (Greiner, Kremsmünster, Austria) at 20 °C, 100 nL of protein mixture were mixed with 100 nL of crystallization buffer using a Mosquito. The crystallization buffer varied for the different proteins, PGRP-LB wild-type (0.1 M HEPES pH 7.0, 30% *v*/*v* Jeffamine^®^ ED-2003), PGRP-LB_C160S (0.2 M Sodium thiocyanate, 20%c *w*/*v* PEG 3350), PGRP-LB_Y78F (0.2 M Sodium chloride, 0.1 M HEPES pH 7.0, 20% *w*/*v* PEG 6000) and PGRP-LB_Y78F + TCT (30% *v*/*v* 2-Propanol, 100 mM Tris base HCl pH 8.5, 30% *w*/*v* PEG 3350). The crystals were cryo-protected adding glycerol to a concentration of 25%. Data were collected on I24 and I04 (Diamond Light Source, Oxfordshire, UK).

### 3.7. Structure Determination

Diffraction data sets were indexed and integrated with DIALS [44]. The resulting integrated data sets were scaled with AIMLESS [45,46]. The structure of the PGRP-LB and its mutants were determined by molecular replacement with PHASER [47] using the previously published PGRP-LB structure (PDB 1OHT) [21]. The molecular replacement models were iteratively rebuilt manually with *Coot* [48] and refined with REFMAC5 [49,50]. Crystallographic data statistics are summarized in Table 2. The figures showing PGRP-LB structural features were displayed with CCP4mg [33].

The X-ray emission fluorescence spectrum were analyzed with pyMCA [34]. The ligand diagram interaction was made using LigPlot+ [35].

### 3.8. 3D Modelling of the Entire TCT

The TCT was build using the geometry of the tetrapeptide from our PGRP-LB_Y78F + TCT structure and the geometry of the sugar from the PGRP-LC and PGRP-LE complexed to TCT structures (PDB 2F2L & 2CB3) [23,24]. The energy was minimized with Maestro and Prime (Schrödinger, LLC, New York, NY, USA, 2020).

## Figures and Tables

**Figure 1 ijms-22-04957-f001:**
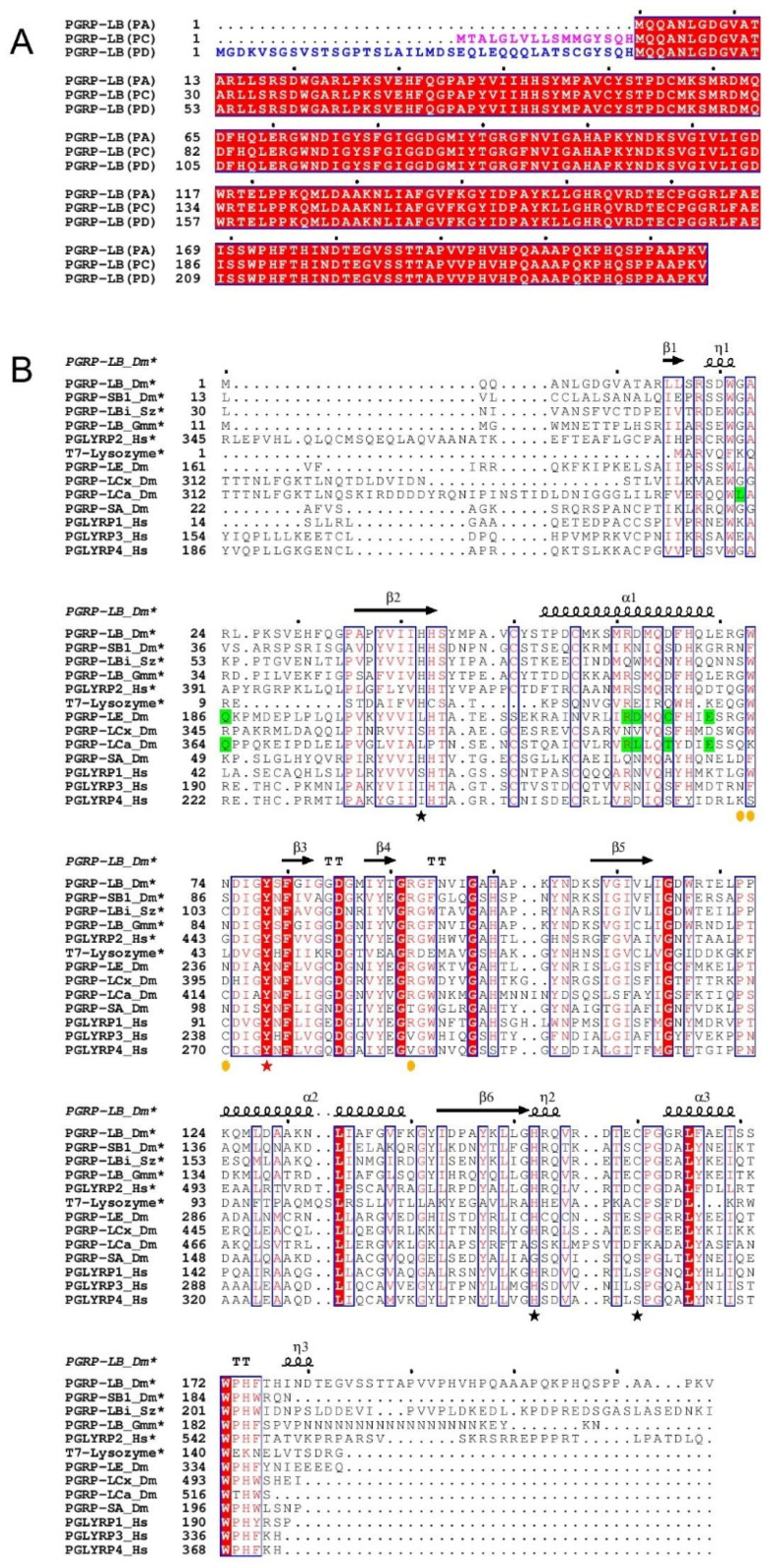
(**A**) Multiple sequence alignment of different isoforms of *Drosophila melanogaster* PGRP-LB. The signal peptide of PGRP-LB^PC^ is highlighted in pink while the N-terminal part of PGRP-LB^PD^ with no homology is highlighted in blue. The fully conserved residues are framed and highlighted in red. (**B**) Multiple sequence alignment of catalytic and non-catalytic PGRPs. The residues indicated by black stars are responsible for the Zn^2+^ interaction in *Drosophila* PGRP-LB (H42, H152 and C160), the residue indicated by red star makes a fourth coordination with Zn^2+^ through a water molecule in *Drosophila* PGRP-LB (Y78). Orange circles indicate the residues responsible for the DAP/Lys specificity. Residues highlighted in green indicate residues responsible for the interaction with the sugar moiety of TCT at the dimer interface in complexed structure PGRP-LC and PGRP-LE. The conserved residues are framed and highlighted in red (fully conserved) or written in red (partially conserved). Only the *Drosophila* PGRP-LB^PA/PC^ isoform is represented in this alignment. Dm = *Drosophila melanogaster*, Hs = *Homo sapiens*, Sz = *Sitophilus zeamais* and Gmm = *Glossina morsitans morsitans*. * for the catalytic proteins. Figures were made using ESPript3.0 [31].

**Figure 2 ijms-22-04957-f002:**
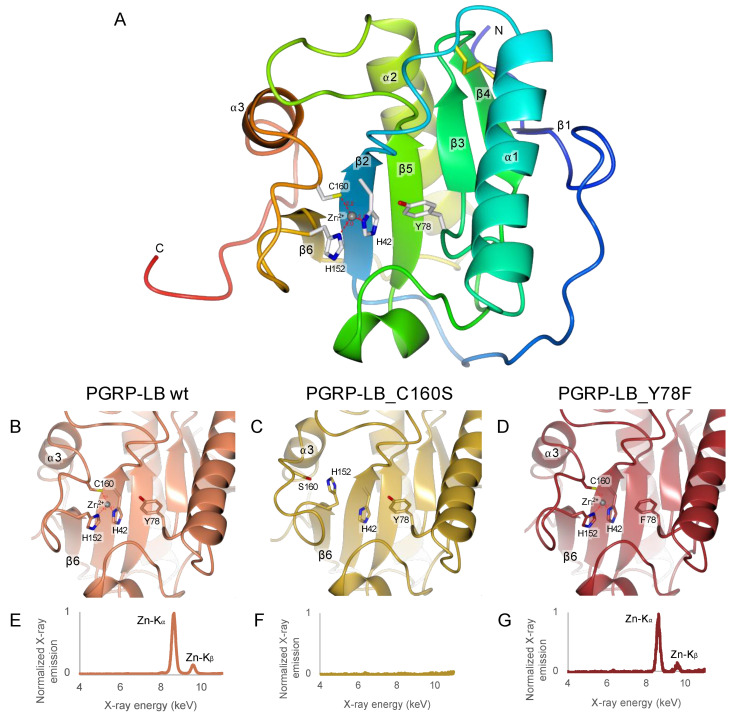
Analysis of Zn^2+^ interaction in *Drosophila* PGRP-LB^PA/PC^ wild-type and its mutants. (**A**) Overall structure of *Drosophila* PGRP-LB^PA/PC^ wild-type (PDB 7NSX). (**B**–**D**) Focus of the X-ray structures of the *Drosophila* PGRP-LB^PA/PC^ wild-type and C160S and Y78F mutants in the Zn^2+^ binding pocket. The C160S mutant lacks Zn^2+^ leading to a RMSD of 2.37 Å on the loop between the sheet β6 and the helix α3. (**E**–**G**) Emission fluorescence spectrum on crystals of the *Drosophila* PGRP-LB^PA/PC^ wild-type and C160S and Y78F mutants. The structure figures were made using CCP4mg [33] and the X-ray emission fluorescence spectrum was analyzed with pyMCA [34].

**Figure 3 ijms-22-04957-f003:**
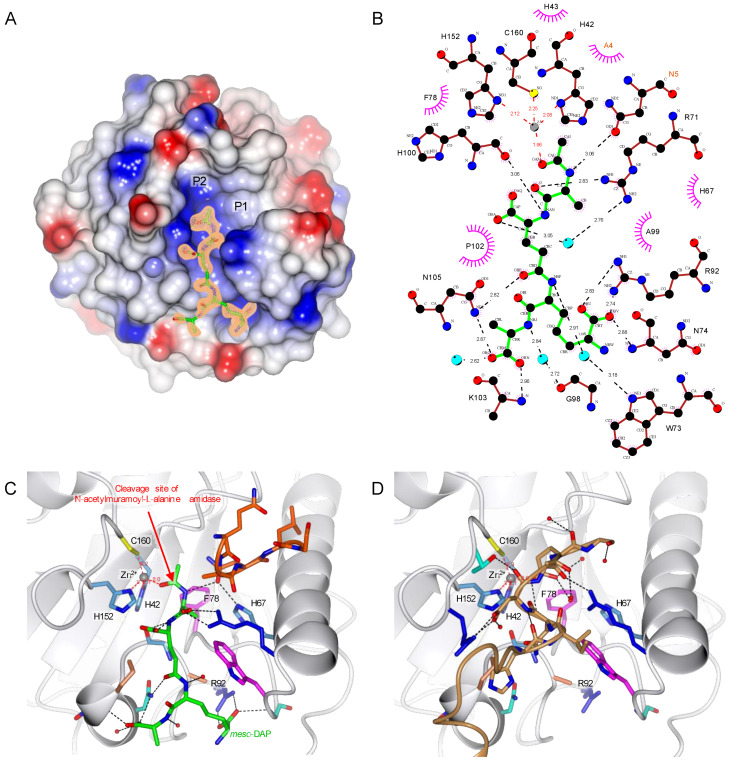
TCT binding to *Drosophila* PGRP-LB^PA/PC^. (**A**) Overall structure of *Drosophila* PGRP-LB^PA/PC^ complexed with TCT (PDB 7NT0). The surface of the protein represents its electrostatic potential, positive potential is represented in blue and negative in red. The TCT is represented in green, and the initial F_o_-F_c_ electron density at 3σ is shown around the TCT. P1 and P2 are possible pockets for the sugar moiety. (**B**) TCT hydrogen bonding network. TCT is represented in green, hydrogen bonds are represented in black dotted lines, red dotted lines show the Zn^2+^ (grey ball) chelation and hydrophobic interactions in the pink circle. The residue belonging to the symmetry-related molecule are annotated in orange. (**C**) Interaction of the TCT within the binding pocket. TCT is shown in green, and the protein is in ribbon representation with side chains of the TCT-interacting residues shown according to their residue property. Hydrogen bonds are represented in black dotted lines; red dotted lines show the Zn^2+^ (grey ball) chelation. The symmetry related chain of 5 residues is colored in orange. The red arrow indicates the cleavage site of the amidase PGRPs (**D**) TCT binding pocket in the apo protein, the symmetry-related molecule represented in brown is filling the site of the TCT. The structure figures were made using CCP4mg [33] and the interaction diagram with LigPlot+ [35].

**Figure 4 ijms-22-04957-f004:**
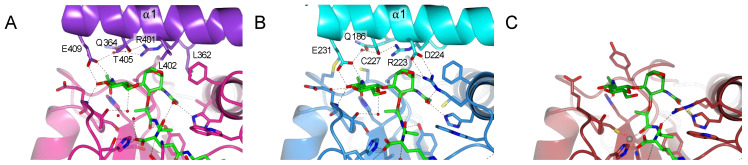
Interaction between carbohydrate moiety of the TCT represented in green and *Drosophila* PGRPs. Only the residues from the helix α1 are responsible for the dimerization and interactions with TCT are annotated. (**A**) Dimer interface of PGRP-LCx in pink and PGRP-LCa in violet (PDB 2F2L) [23]. (**B**) Dimer interface of PGRP-LE, the main interacting chain is in blue and the chain at the dimer interface is in cyan (PDB 2CB3) [24]. (**C**) Model for the interaction of the PGRP-LB^PA/PC^ with the sugar moiety of the TCT. The missing sugar moiety has been modelled using Maestro and Prime (Schrödinger, LLC, New York, NY, 2020) and colored in darker green. Hydrogen bonds are represented by black dotted lines; red dotted lines show the Zn^2+^ (grey ball) chelation. The structure figures were made using CCP4mg [33].

**Figure 5 ijms-22-04957-f005:**
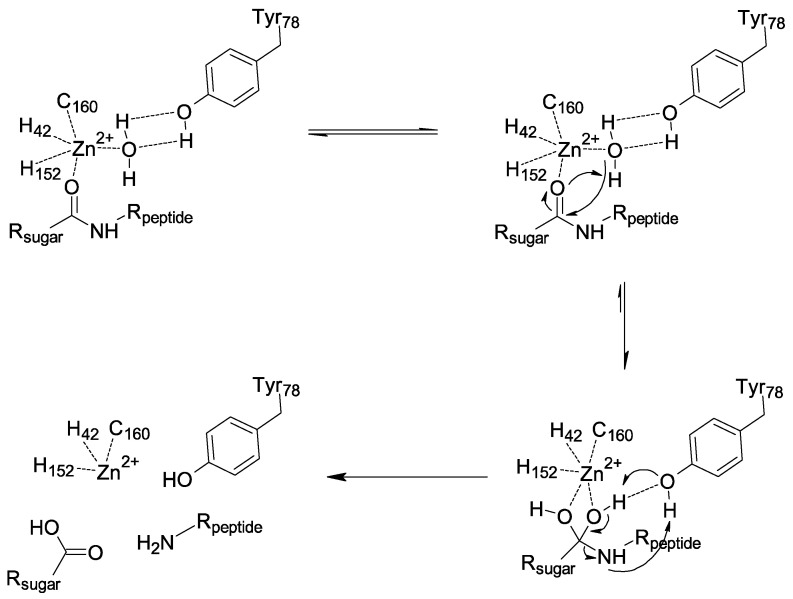
Proposed mechanism for *Drosophila* PGRP-LB.

**Table 1 ijms-22-04957-t001:** In vitro enzymatic activity of *Drosophila* PGRP-LB isoforms and their mutants toward different substrates.

Specific Activity (nmol.min^−1^.mg of Protein^−1^)									
Protein	Substrate								
	*E. coli* PGN (DAP Type)	GM(anh)- tetra_DAP_ (TCT)	GM(anh)- tetra_DAP_ dimer *	GM(anh)- tetra_DAP_ dimer **	GM(anh)-penta_Lys_	GM-tetra_DAP_	M(anh)- tetra_DAP_	M-tetra_DAP_	GM(anh)- tetra_DAP_ (No Zn^2+^)
PGRP-LB^PA/PC^	1570 ± 40	4525 ± 682	1807 ± 341	992 ± 671	167 ± 34	1132 ± 230	1456 ± 542	669 ± 215	3468 ± 624
PGRP-LB^PD^	660 ± 27	3730 ± 597	1055 ± 220	330 ± 312	126 ± 28	832 ± 279	1441 ± 235	266 ± 84	4908 ± 1351
PGRP-LB^PA/PC^_H42A	NA	13 ± 4	11 ± 2	NA	NA	15 ± 6	NA	NA	10 ± 4
PGRP-LB^PA/PC^_Y78F	NA	NA	NA	NA	NA	NA	NA	NA	NA
PGRP-LB^PA/PC^_H152A	NA	7 ± 2	4 ± 1	NA	NA	NA	NA	NA	7 ± 3
PGRP-LB^PA/PC^_C160S	NA	26 ± 5	28 ± 8	7 ± 9	NA	NA	10 ± 4	NA	NA

The chemical structures of the different substrates are represented in Appendix A. NA indicates no activity. * Only one of the two sugars is removed, ** both sugars are removed.

**Table 2 ijms-22-04957-t002:** Data collection and structure refinement statistics.

	PGRP-LB^PA/PC^	PGRP-LB^PA/PC^_C160S	PGRP-LB^PA/PC^_Y78F	PGRP-LB^PA/PC^_Y78F + TCT
PDB code	7NSX	7NSY	7NSZ	7NT0
**Data collection**				
Space Group	C222_1_	P1	P6_1_22	P1
Cell parameters				
a, b, c (Å)	39.87, 70.55, 112.83	41.02, 49.19, 52.14	40.50, 40.50, 338.94	38.59, 52.17, 55.58
α, β, γ (°)	90, 90, 90	71.82, 79, 66.84	90, 90, 120	92.11, 105.12, 111.51
Resolution range	56.48–1.90 (1.94–1.90)	49.39–1.40 (1.40–1.42)	56.55–1.30 (1.32–1.30)	53.07–1.80 (1.84–1.80)
Unique reflection	12,942 (808)	67,209 (3218)	42,802 (1995)	34,852 (2038)
R_meas_	0.154 (1.406)	0.043 (0.420)	0.099 (0.476)	0.099 (0.564)
Completeness (%)	99.8 (99.2)	96.1 (92.2)	99.9 (98.9)	97.7 (97.2)
Multiplicity	7.9 (7.2)	3.4 (2.6)	31.5 (13.0)	6.9 (6.8)
I/σ(I)	8.5 (1.4)	14.1 (2.5)	20.8 (2.8)	11.9 (3.3)
CC1/2	0.993 (0.562)	0.999 (0.875)	0.998 (0 0.955)	0.998 (0.891)
**Refinement**				
R_work_/R_free_	0.180/0.213	0.167/0.188	0.186/0.221	0.145/0.180
RMSD				
Bond length (Å)	0.0098	0.0145	0.0143	0.0131
Bond angle (°)	1.597	1.906	1.904	1.684
B factors (Å^2^)				
Protein	29.09	18.05	14.32	20.44
Ion	19.64	-	11.35	14.68
Ligand	-	-	-	25.35
Water	33.55	30.97	26.97	30.78
Ramachandran (%)				
Favored	94.64	94.22	94.71	93.96
Allowed	4.17	4.56	4.71	5.44
Outliers	1.19	1.22	0.59	0.60

Statistics for the highest-resolution shell are shown in parentheses.

## Data Availability

The atomic coordinates and structural factors for the crystal structures in this paper have been deposited to the Protein Data Bank in Europe (PDBe) with the accession numbers 7NSX (PGRP-LB^PA/PC^), 7NSY (PGRP-LB^PA/PC^_C160S), 7NSZ (PGRP-LB^PA/PC^_Y78F) and 7NT0 (PGRP-LB^PA/PC^_Y78F + TCT).

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
