# Peer review of "PGRP-LB: An Inside View into the Mechanism of the Amidase Reaction"

_ijms, 2021, doi:10.3390/ijms22094957_

Round 1

Reviewer 1 Report

Current version of the manucript is well organized and written. The authors have found that Drosophila PGRP-LB belongs to amidase PGRPs and 20 downregulates the immune deficiency (IMD) pathway by cleaving meso-2,6-diaminopimelic (meso- 21 DAP or DAP)-type PGN.  In particular, they found out that the specific activity of different isoforms 24 of Drosophila PGRP-LB towards various PGN substrates to understand their specificity and role in 25 Drosophila immunity.

I am sure that quality of the manuscript is good enough to be accepted at IJMS.

Author Response

On behalf of all authors, I want to thank the expert reviewer for the careful and relevant reading of entitled “PGRP-LB: an inside view into the mechanism of the amidase reaction”.

As requested, please find below our point-by-point answers to the remarks made by the reviewer.

(x) English language and style are fine/minor spell check required

Adam Crawshaw, a native English speaker checks the grammar, spelling, punctuation and phrasing of our paper. A. Crawshaw was thanked for his proof‑reading of the paper at the acknowledgment section line 470.

Reviewer 2 Report

The article analyzed the structural features and substrate specificity of three isozymes of Drosophila peptidoglycan amidase that cleaves the meso-DAP-containing side chains of peptidoglycan, which leads to the downregulation of the immune deficiency pathway of Drosophila. Through the analysis of various peptidoglycan substrates, the authors showed that three amidase isozymes exhibit the similar substrate specificity. By the structural analysis and site-directed mutagenesis, the importance of an unexpected residue, Y78, was identified. The Y78F mutation completely abolished the activity of peptidoglycan amidase. Based on these data, the authors suggested the possible role of the Y78 residue in the amidase reaction. Several points should be improved.

  1. The quality of Fig. 1B should be improved. e.g. There are inaccurate positions of boxes.

  1. ND (not determined) should be changed to NA or specific parameters, especially in Y78F of GM(anh-tetraDAP(no Zn2+) substrate.

  1. The quality (SDS-PAGE gel) of all purified proteins in Table 1 should be shown in the supplementary data.

  1. Fig. 3C: TCT is not N-acetylmuramyoyl-L-alanine amidase.

Author Response

On behalf of all authors, I want to thank the expert reviewer for the careful and relevant reading of entitled “PGRP-LB: an inside view into the mechanism of the amidase reaction”.

As requested, please find below our point-by-point answers to the remarks made by the reviewer.

1. The quality of Fig. 1B should be improved. e.g. There are inaccurate positions of boxes.

The quality of Fig. 1B has been improved and Fig. 1B position of boxes corrected in the revised version of the manuscript.

2. ND (not determined) should be changed to NA or specific parameters, especially in Y78F of GM(anh-tetraDAP(no Zn2+) substrate.

Initially, we did not measure the activities for the mutation Y78F and for the isoform PGRP‑LBPD without Zn2+ in the buffer as we wanted first to measure the only effect on the chelating Zn2+ residues (H42, H152 and C160) and compared it to the WT isoform. According reviewer suggestion, we have measured amidase activities of PGRP-LBPA/PC_Y78F and PGRP-LBPD toward TCT without Zn2+ in the buffer. ND (Not determined) has been changed to NA (table 1) for PGRP‑LBPA/PC_Y78F as no activity has been observed. Only the Y78F mutation caused complete loss of the enzymatic activity for every compounds with or without Zn2+ in the buffer. About PGRP-LBPD amidase activity toward TCT without Zn2+ in the buffer, ND has been changed to specific activity value 4908 ± 1351 nmol.min-1.mg of protein-1 (Table 1) confirming even without any Zn2+ in the media, the activity remains the same confirming the strong chelation of Zn2+ by the protein as we have already observed in wild-type PGRP-LBPA/PC isoform. These new results have been added in the revised manuscript and accordingly the text has been changed lines 229-231.

3. The quality (SDS-PAGE gel) of all purified proteins in Table 1 should be shown in the supplementary data.

The quality (SDS-PAGE gel) of all purified proteins in Table 1 has been added in the supplementary data as figure S4 and the following sentence has been added “The protein purification quality was assessed by SDS-PAGE (Figure S4)” line 392 in the revised manuscript.

4. Fig. 3C: TCT is not N-acetylmuramyoyl-L-alanine amidase.

The arrow indicates in FIG. 3C the cleavage site of the enzyme on the TCT and not the enzyme N-acetylmuramyoyl-L-alanine amidase. The text has been replaced in Figure 3C with “Cleavage site of N-acetylmuramyoyl-L-alanine amidase” and the following sentence “The red arrow indicates the cleavage site of the amidase PGRPs” in figure 3C legend line 270 has been added in the revised manuscript.

Round 2

Reviewer 2 Report

.